# A Message Passing Perspective on Learning Dynamics of Contrastive Learning

**Yifei Wang**[1][*] **Qi Zhang**[2][*] **Tianqi Du**[1] **Jiansheng Yang**[1] **Zhouchen Lin**[2,3,4] **Yisen Wang**[2,3][†]

[1]School of Mathematical Sciences, Peking University
[2]National Key Lab of General Artificial Intelligence,
  School of Intelligence Science and Technology, Peking University
[3]Institute for Artificial Intelligence, Peking University
[4]Peng Cheng Laboratory

## Abstract

In recent years, contrastive learning achieves impressive results on self-supervised visual representation learning, but there still lacks a rigorous understanding of its learning dynamics. In this paper, we show that if we cast a contrastive objective equivalently into the feature space, then its learning dynamics admits an interpretable form. Specifically, we show that its gradient descent corresponds to a specific message passing scheme on the corresponding augmentation graph. Based on this perspective, we theoretically characterize how contrastive learning gradually learns discriminative features with the alignment update and the uniformity update. Meanwhile, this perspective also establishes an intriguing connection between contrastive learning and Message Passing Graph Neural Networks (MP-GNNs). This connection not only provides a unified understanding of many techniques independently developed in each community, but also enables us to borrow techniques from MP-GNNs to design new contrastive learning variants, such as graph attention, graph rewiring, jumpy knowledge techniques, etc. We believe that our message passing perspective not only provides a new theoretical understanding of contrastive learning dynamics, but also bridges the two seemingly independent areas together, which could inspire more interleaving studies to benefit from each other. The code is available at `https://github.com/PKU-ML/Message-Passing-Contrastive-Learning`.

## 1 Introduction

Contrastive Learning (CL) has become arguably the most effective approach to learning visual representations from unlabeled data (Chen et al., 2020b; He et al., 2020; Chen et al., 2020c; Wang et al., 2021a; Chen et al., 2020d; 2021; Caron et al., 2021). However, till now, we actually know little about how CL gradually learns meaningful features from unlabeled data. Recently, there has been a burst of interest in the theory of CL. However, despite the remarkable progress that has been made, existing theories of CL are established for either an *arbitrary* function $f$ in the function class $\mathcal{F}$ (Saunshi et al., 2019; Wang et al., 2022) or the optimal $f^*$ with minimal contrastive loss (Wang & Isola, 2020; HaoChen et al., 2021; Wang et al., 2022). Instead, a theoretical characterization of the learning dynamics is largely overlooked, which is the focus of this work.

Perhaps surprisingly, we find out that the optimization dynamics of contrastive learning corresponds to a specific message passing scheme among different samples. Specifically, based on a reformulation of the alignment and uniformity losses of the contrastive loss into the feature space, we show that the derived alignment and uniformity updates actually correspond to message passing on two different graphs: the alignment update on the augmentation graph defined by data augmentations, and the uniformity update on the affinity graph defined by feature similarities. Therefore, the combined contrastive update is a competition between two message passing rules. Based on this perspective, we further show that the equilibrium of contrastive learning can be achieved when the two message rules are balanced, *i.e.,* when the learned distribution $\mathcal{P}_\theta$ matches the ground-truth data distribution $\mathcal{P}_d$, which provides a clear picture for understanding the dynamics of contrastive learning.

---

[*]Equal Contribution.
[†]Corresponding Author: Yisen Wang (yisen.wang@pku.edu.cn).

Meanwhile, as message passing is a general paradigm in many scenarios, the message passing perspective of contrastive learning above also allows us to establish some intriguing connections to these seemingly different areas. One particular example is in graph representation learning. Message Passing Graph Neural Networks (MP-GNNs) are the prevailing designs in modern Graph Neural Networks (GNNs), including numerous variants like GCN (Kipf & Welling, 2017), GAT (Veličković et al., 2018), and even the Transformers (Vaswani et al., 2017). There is a vast literature studying its diffusion dynamics and representation power (Li et al., 2018; Oono & Suzuki, 2020; Wang et al., 2021b; Li et al., 2022; Dong et al., 2021; Xu et al., 2019; 2018; Chen et al., 2022). Therefore, establishing a connection between contrastive learning (CL) and MP-GNNs will hopefully bring new theoretical and empirical insights for understanding and designing contrastive learning methods. In this work, we illustrate this benefit from three aspects: 1) we establish formal connections between the basic message passing mechanisms in two domains; 2) based on this connection, we discover some close analogies among the representative techniques independently developed in each domain; and 3) borrowing techniques from MP-GNNs, we design two new contrastive learning variants, and demonstrate their effectiveness on benchmark datasets. We summarize our contributions as follows:

- **Learning Dynamics.** We reformulate of the contrastive learning into the feature space and develop a new decomposition of the alignment and uniformity loss. Based on this framework, we show that the alignment and uniformity updates correspond to two different message passing schemes, and characterize the equilibrium states under the combined update. This message perspective provides a new understanding of contrastive learning dynamics.

- **Connecting CL and MP-GNNs.** Through the message passing perspective of contrastive learning (CL), we show that we can establish an intriguing connection between CL and MP-GNNs. We not only formally establish the equivalence between alignment update and graph convolution, uniformity update and self-attention, but also point out the inherent analogies between important techniques independently developed in each domain.

- **New Designs Inspired by MP-GNNs.** We also demonstrate the empirical benefits of this connection by designing two new contrastive learning variants borrowing techniques from MP-GNNs: one is to avoid the feature collapse of alignment update by multi-stage aggregation, and one is to adaptively align different positive samples with by incorporating the attention mechanism. Empirically, we show that both techniques leads to clear benefits on benchmark datasets. In turn, their empirical successes also help verify the validness of our established connection between CL and MP-GNNs.

## 2  A MESSAGE PASSING PERSPECTIVE ON CONTRASTIVE LEARNING

In this section, we develop a message passing perspective for understanding the dynamics of contrastive learning. We begin by reformulating the contrastive loss into the feature space with a new decomposition. We then study the update rules derived from the alignment and uniformity losses, and explain their behaviors from a message passing perspective. When combined together, we also characterize how the two updates strike a balance at the equilibrium states. And we finish this section with a proof-of-idea to illustrate the effectiveness of our derive message passing rules.

### 2.1  BACKGROUND, REFORMULATION, AND DECOMPOSITION

We begin our discussion by introducing the canonical formulation of contrastive learning methods in the parameter space, and present their equivalent formulation in the feature space.

**Contrastive Learning (CL).** Given two positive samples $(x, x^+)$ generated by data augmentations, and an independently sampled negative sample $x'$, we can learn an encoder $f_\theta : \mathbb{R}^d \to \mathbb{R}^m$ with the wide adopted InfoNCE loss (Oord et al., 2018):

$$\mathcal{L}_{\text{nce}}(\theta) = -\mathbb{E}_{x,x^+}[f_\theta(x)^\top f_\theta(x^+)] + \mathbb{E}_x \log \mathbb{E}_{x'}[\exp(f_\theta(x)^\top f_\theta(x'))], \qquad (1)$$

where the form term pulls positive samples $(x, x^+)$ together by encouraging their similarity, and the latter term pushes negative pairs $(x, x')$ apart. In practice, we typically randomly draw $M$ negative samples to approximate the second term.

In contrastive learning, the encoder is parameterized by deep neural networks, making it hardly amenable for formal analysis. Wen & Li (2021) resort to single-layer networks with strong assumptions on data distribution, but it is far from practice. Instead, in this work, we focus on the dynamics

of the learned *data features* $f_\theta(x)$ in the feature space $\mathcal{R}^m$. As over-parameterized neural networks are very expressive and adaptive, we assume the feature matrix $F_\theta$ to be *unconstrained* (updated freely ignoring dependences on parameters and sample complexity) as in HaoChen et al. (2021). This assumption is also widely adopted in deep learning theory, such as, layer peeled models (Mixon et al., 2022; Zhu et al., 2021). The unconstrained feature assumption enables us to focus on the relationship between augmented samples. As pointed out by Wang et al. (2022), the support overlap between augmented data is the key for contrastive learning to generalize to downstream tasks. However, we still do not have a clear view of how data augmentations affect feature learning, which is the focus of this work. For a formal exposition, we model data augmentations through the language of augmentation graph.

Following HaoChen et al. (2021), we assume a finite sample space $\mathcal{X}$ (can be exponentially large). Given a set of natural data as $\bar{\mathcal{X}} = \{\bar{x} \mid \bar{x} \in \mathbb{R}^d\}$, we first draw a natural example $\bar{x} \in \mathcal{P}_d(\bar{\mathcal{X}})$, and then draw a pair of samples $x, x^+$ from a random augmentation of $\bar{x}$ with distribution $\mathcal{A}(\cdot|\bar{x})$. We denote $\mathcal{X}$ as the collection of augmented "views", *i.e.*, $\mathcal{X} = \cup_{\bar{x} \in \bar{\mathcal{X}}} \mathrm{supp}\,(\mathcal{A}(\cdot|\bar{x}))$. Regarding the $N$ samples in $\mathcal{X}$ as $N$ nodes, we can construct an augmentation graph $\mathcal{G} = (\mathcal{X}, A)$ with an adjacency matrix $A$ representing the mutual connectivity. The edge weight $A_{xx'} \geq 0$ between $x$ and $x'$ is defined as their joint probability $A_{xx'} = \mathcal{P}_d(x, x') = \mathbb{E}_{\bar{x}} \mathcal{A}(x|\bar{x}) \mathcal{A}(x'|\bar{x})$. The normalized adjacency matrix is $\bar{A} = D^{-1/2} A D^{-1/2}$, where $D = \deg(A)$ denotes the diagonal degree matrix, *i.e.*, $D_{xx} = w_x = \sum_{x'} A_{xx'}$. The symmetrically normalized Laplacian matrix $L = I - \bar{A}$ is known to be positive semi-definite (Chung, 1997). As natural data and data augmentations are uniformly sampled in practice (Chen et al., 2020b), we assume uniform distribution $w_x = 1/N$ for simplicity.

**New Alignment and Uniformity Losses in Feature Space.** Here, we reformulate the contrastive losses into the feature space, and decompose them into two parts, the alignment loss for positive pairs, and the uniformity loss for negative pairs ($\mathcal{L}_{\mathrm{unif}}$ for InfoNCE loss, and $\mathcal{L}_{\mathrm{unif}}^{(\mathrm{sp})}$ for spectral loss):

$$\mathcal{L}_{\mathrm{align}}(F) = \mathrm{Tr}(F^\top L F) = \frac{1}{2}\mathbb{E}_{x,x^+}\|f(x) - f(x^+)\|^2, \tag{2a}$$

$$\mathcal{L}_{\mathrm{unif}}(F) = \mathrm{LSE}(FF^\top) - \|F\|^2 = \mathbb{E}_x \log \mathbb{E}_{x'} \exp(f(x)^\top f(x')) - \mathbb{E}_x \|f(x)\|^2, \tag{2b}$$

where $\mathrm{LSE}(X) = \mathrm{Tr}(D \log(\deg(D \exp(D^{-1/2} X D^{-1/2}))))$ serve as a pseudo "norm".[1] The following proposition establishes the equivalence between original loss and our reformulated one.

**Proposition 1.** *When the $x$-th row of $F$ is defined as $F_x = \sqrt{w_x} f_\theta(x)$, we have:*

$$\mathcal{L}_{\mathrm{nce}}(\theta) = \mathcal{L}_{\mathrm{align}}(F_\theta) + \mathcal{L}_{\mathrm{unif}}(F_\theta). \tag{3}$$

**Remark on Loss Decomposition.** Notably, our decomposition of contrastive loss is different from the canonical decomposition proposed by Wang & Isola (2020). Taking the InfoNCE loss as an example, they directly take the two terms of Eq. 1 as the alignment and uniformity losses, which, however, leads to two improper implications without feature normalization. First, minimizing their alignment loss $\mathcal{L}_{\mathrm{align}}(\theta) = -\mathbb{E}_{x,x^+} f_\theta(x)^\top f_\theta(x^+)$ alone is an ill-posed problem, as it approaches $-\infty$ when we simply scale the norm of $F$. In this case, the alignment loss gets smaller while the distance between positive pairs gets larger. Second, minimizing the uniformity loss alone will bridge the distance closer. Instead, with our decomposition, the alignment loss (Eq. 2a) is positive definite and will bring samples together with smaller loss, and a larger uniformity loss (Eq. 2b) will expand their distances (Figure 1). Thus, our decomposition in Eq. 2 seems more properly defined than theirs.

Based on the reformulation in Eq. 2, we will study the dynamics of each objective in Section 2.2.1 and give a unified analysis of their equilibrium in Section 2.3, both from a message passing perspective.

## 2.2 A Message Passing Perspective on Alignment and Uniformity

### 2.2.1 Alignment Update is Message Passing on Augmentation Graph

In fact, the reformulated alignment loss $\mathcal{L}_{\mathrm{align}}(F) = \mathrm{Tr}(F^\top L F)$ is widely known as the *Laplacian regularization* that has wide applications in various graph-related scenarios (Ng et al., 2001; Ando & Zhang, 2006). Its gradient descent with step size $\alpha > 0$ gives the following update rule:

$$\text{(global update)} \quad F^{(t+1)} = F^{(t)} - \alpha \nabla_{F^{(t)}} \mathcal{L}_{\mathrm{align}}(F^{(t)}) = \left[(1 - 2\alpha)I + 2\alpha \bar{A}\right] F^{(t)}. \tag{4}$$

---

[1] Here $\log, \exp$ are element-wise operations.

One familiar with graphs can immediately recognize that this update rule is a matrix-form message passing scheme of features $F^{(t)}$ with a propagation matrix $P = (1 - 2\alpha)I + 2\alpha\bar{A}$. For those less familiar with the graph literature, let us take a look at the *local* update of the feature $F_x$ of each $x$:

$$\text{(local update)} \quad F_x^{(t+1)} = (1 - 2\alpha)F_x^{(t+1)} + 2\alpha\sum_{x'\in\mathcal{N}_x}\bar{A}_{xx'}F_{x'}^{(t+1)}. \quad (5)$$

We can see that the alignment loss updates each feature $F_x$ by a weighted sum that aggregates features from its neighborhood $\mathcal{N}_x = \{x' \in \mathcal{X} | \bar{A}_{xx'} > 0\}$. In the context of contrastive learning, this reveals that the alignment loss is to propagate features among positive pairs, and in each step, more alike positive pairs $x, x^+$ (with higher edge weights $A_{xx^+}$) will exchange more information.

Next, let us turn our focus to how this message passing scheme affects downstream classification. As for contrastive learning, because positive samples come from the same class, this alignment loss will help cluster same-class samples together. However, it does not mean that any augmentations can attain a minimal intra-class variance. Below, we formally characterize the effect of data augmentations on the feature clustering of each class $k$. We measure the degree of clustering by the distance between all intra-class features to a one-dimensional subspace $\mathcal{M}$.

**Proposition 2.** *Denote the distance between a feature matrix $X$ and a subspace $\Omega$ by $d_\Omega(X) := \inf\{\|X - Y\|_F \mid Y \in \Omega\}$. Assume that data augmentations are label-preserving. Then, for samples in each class $k$, denoting their adjacency matrix as $\bar{A}_k$ and their features at $t$-th step as $F_k^{(t)}$, the message passing in Eq. 4 leads to a smaller or equal distance to an one-dimensional subspace $\mathcal{M}$*

$$d_\mathcal{M}(F_k^{(t+1)}) \leq |1 - 2\alpha\lambda_{\mathcal{G}_k}| \cdot d_\mathcal{M}(F_k^{(t)}), \quad (6)$$

*where $\lambda_{\mathcal{G}_k} \in [0, 2]$ denotes the intra-class algebraic connectivity of $\mathcal{G}$ (Chung, 1997), and $\mathcal{M} = \{e_1 \otimes y | y \in \mathbb{R}^c\}$, and $e_1$ is the eigenvector corresponding to the largest eigenvalue of $\bar{A}_k$.*

An important message of Proposition 2 is that the alignment update does *not* necessarily bring better clustering, which depends on the algebraic connectivity $\lambda_{\mathcal{G}_k}$: 1) only a non-zero $\lambda_{\mathcal{G}_k}$ guarantees the strict decrease of intra-class distance, and 2) a larger $\lambda_{\mathcal{G}_k}$ indicates an even faster decrease. According to spectral graph theory, $\lambda_{\mathcal{G}_k}$ is the second-smallest eigenvalue of the Laplacian $L$, and its magnitude reflects how connected the graph is. In particular, $\lambda_{\mathcal{G}_k} > 0$ holds if and only if the intra-class graph is connected (Chung, 1997). Indeed, when the class is not connected, the alignment update can never bridge the disjoint components together. While in contrastive learning, the design of data augmentations determines $\mathcal{G}_k$ as well as $\lambda_{\mathcal{G}_k}$. Therefore, our message passing analysis *quantitatively* characterize how augmentations affect feature clustering in the contrastive learning dynamics.

### 2.2.2 UNIFORMITY UPDATE IS MESSAGE PASSING ON THE AFFINITY GRAPH

Contrary to the alignment loss, the uniformity loss instead penalizes the similarity between every sample pair $x, x' \in \mathcal{X}$ to encourage feature uniformity. Perhaps surprisingly, from a feature space perspective, its gradient descent rule, namely the uniformity update, also corresponds to a message passing scheme with the affinity graph $\mathcal{G}'$ with a "fake" adjacency matrix $A'$:

$$\text{(global update)} \quad F^{(t+1)} = F^{(t)} - \alpha\nabla_{F^{(t)}}\mathcal{L}_{\text{unif}}(F^{(t)}) = [(1 + 2\alpha)I - 2\alpha\bar{A}'_{\text{sym}}]F^{(t)}, \quad (7)$$

where $\bar{A}'_{\text{sym}} = D'_{\text{exp}}{}^{-1}A'_{\text{exp}} + A'_{\text{exp}}D'_{\text{exp}}{}^{-1}$, and $A'_{\text{exp}} = \exp(D^{-1/2}F^{(t)}F^{(t)\top}D^{-1/2})$, $D'_{\text{exp}} = \deg(A'_{\text{exp}})$. When we further apply stop gradient to the target branch (as adopted in BYOL (Grill et al., 2020) and SimSiam (Chen & He, 2021)), we can further simplify it to

$$\text{(global update with stop gradient)} \quad F^{(t+1)} = [(1+\alpha)I - \alpha\bar{A}']F^{(t)}, \text{ where } \bar{A}' = D'_{\text{exp}}{}^{-1}A'_{\text{exp}}. \quad (8)$$

We can also interpret the uniformity update as a specific message passing scheme on the *affinity graph* $\mathcal{G}' = (\mathcal{X}, A')$ with node set $\mathcal{X}$ and adjacency matrix $A'$. In comparison to the augmentation graph $\mathcal{G}$ that defines the edge weights via data augmentations, the affinity graph is instead constructed through the estimated *feature similarity* between $(x, x')$. The contrastive learning with two graphs resembles the wake-sleep algorithm in classical unsupervised learning (Hinton et al., 1995).

Further, we notice that the affinity matrix $\bar{A}'$ is induced by the RBF kernel $k_{\text{RBF}}(u, v) = \exp(-\|u - v\|^2/2)$ when the features are normalized, *i.e.,* $\|f(x)\| = 1$ (often adopted in practice (Chen et al., 2020b)). Generally speaking, the RBF kernel is more preferable than the linear kernel as it is more non-Euclidean and non-parametric. Besides, because the message passing in the alignment update utilizes $\bar{A}$, which is also non-negative and normalized, the InfoNCE uniformity update with $\bar{A}'_{\text{nce}}$ can be more stable and require less hyperparameter tuning to cooperate with the alignment update. Thus, our message passing view not only provides a new understanding of the uniformity update, but also helps explain the popularity of the InfoNCE loss.

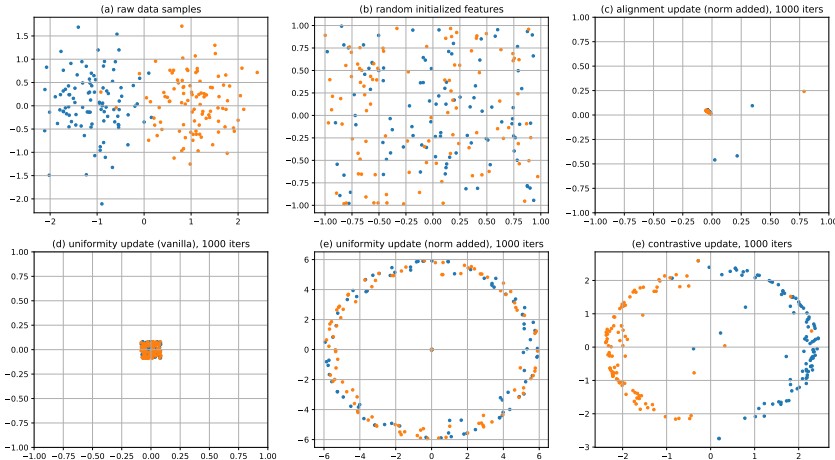

Figure 1: Mimicking contrastive learning with our message passing rules on synthetic data. The data samples $\mathcal{D}$ are generated following the isotropic Gaussian distribution with means $(-1, 0)$ and $(1, 0)$ (two classes, blue and yellow), and variance $0.7$. We then construct an augmentation graph $\mathcal{G}$ by drawing edged $A_{ij} = 1$ for any data pair satisfying $\|x_i - x_j\|_2 \leq 0.4$. For mimicking contrastive learning, we firstly initialize all features $\{f_i\}$ uniformly in $[-1, 1]^2$, and then perform message passing of these features using $\mathcal{G}$ for $1,000$ steps with step size $0.1$. Best viewed in color.

## 2.3  THE EQUILIBRIUM BETWEEN THE ALIGNMENT AND UNIFORMITY UPDATE

Combining the alignment update and the uniformity update discussed above, we arrive at our final update rule for contrastive learning with the (regularized) InfoNCE loss (with stop gradient):

$$F^{(t+1)} = F^{(t)} - \alpha \nabla_{F^{(t)}} \mathcal{L}_{\text{nce}}(F^{(t)}) = F^{(t)} + \alpha(\bar{A} - \bar{A}')F^{(t)}, \tag{9}$$

which is a special message passing scheme using the difference between the augmentation (data) and affinity (estimated) adjacency matrices $A_{\text{diff}} = \bar{A} - \bar{A}'$. Accordingly, we can reformulate the contrastive loss more intuitively as:

$$\mathcal{L}(F) = -\operatorname{Tr}(\text{sg}(A_{\text{diff}})FF^\top) = -\operatorname{Tr}(\bar{A}FF^\top) + \operatorname{Tr}(\text{sg}(A')FF^\top) = \mathcal{L}_{\text{align}}(F) + \mathcal{L}_{\text{unif}}(F), \tag{10}$$

where $\text{sg}(\cdot)$ means the stop gradient operator.

**Equilibrium Distribution.** The reformulation above (Eq. 10) also reveals the stationary point of the learning dynamics, that is, $\bar{A}_{\text{diff}} = \bar{A} - A' = 0$, which implies the model equilibrium as follows.

**Proposition 3.** *The contrastive learning dynamics with the InfoNCE loss saturates when the real and the fake adjacency matrices agree, i.e., $\bar{A} = \bar{A}'_{\text{nce}}$, which means that element-wisely, we have:*

$$\forall\, x, x^+ \in \mathcal{X}, \ \ \mathcal{P}_d(x^+|x) = \mathcal{P}_\theta(x^+|x) \triangleq \frac{\exp(f_\theta(x)^\top f_\theta(x^+))}{\sum_{x'} \exp(f_\theta(x)^\top f_\theta(x'))}. \tag{11}$$

*That is, the ground-truth data conditional distribution $\mathcal{P}_d(x^+|x)$ equals to the Boltzmann distribution $\mathcal{P}_\theta(x^+|x)$ estimated from the encoder features.*

In this way, our analysis reveals that contrastive learning (with InfoNCE loss) implicitly learns a probabilistic model $\mathcal{P}_\theta(x^+|x)$ (Eq. 11), and its goal is to approximate the conditional data distribution $P_d(x^+|x)$ defined via the augmentation graph $\mathcal{G}$. This explains the effectiveness of contrastive learning on out-of-distribution detection tasks (Winkens et al., 2020) that rely on good density estimation. In practical implementations of InfoNCE, *e.g.,* SimCLR, a temperature scalar $\tau$ is often introduced to rescale the cosine similarities. With a flexible choice of $\tau$, the model distribution $\mathcal{P}_\theta$ could better approximate sharp (like one-hot) data distribution $\mathcal{P}_d$. A detailed comparison to related work is included in Appendix A.

## 2.4  A PROOF-OF-IDEA EXPERIMENT

At last, we verify the derived three message passing rules (alignment, uniformity, contrastive) through a synthetic experiment using on a synthetic augmentation graph to mimic contrastive learning in the feature space. From Figure 1, we can see that the results generally align well with our theory. First, the alignment update (Section 2.2.1) indeed bridges intra-class samples closer, but there is also

a risk of feature collapse. Second, the uniformity update (Section 2.2.2) indeed keeps all samples uniformly distributed. Third, by combining the alignment and uniformity updates, the contrastive update (Section 2.3) can successfully cluster intra-class samples together while keeping inter-class samples apart, even after long iterations. In this way, we show that like the actual learning behaviors of contrastive learning, the derived contrastive message passing rules can indeed learn stable and discriminative features from unlabeled data.

# 3    CONTRASTIVE LEARNING (CL) AND MESSAGE PASSING GRAPH NEURAL NETWORKS (MP-GNNs): CONNECTIONS, ANALOGIES, AND NEW DESIGNS

In the previous section, we have shown the contrastive learning dynamics corresponds nicely to a message passing scheme on two different graphs, $\bar{A}$ and $\bar{A}'$. In a wider context, message passing is a general technique with broad applications in graph theory, Bayesian inference, graph embedding, etc. As a result, our interpretation of contrastive learning from a message passing perspective enables us to establish some interesting connections between contrastive learning and these different areas.

A noticeable application of message passing is graph representation learning. In recent years, Message Passing Graph Neural Networks (MP-GNNs) become the *de facto* paradigm for GNNs. Various techniques have been explored in terms of different kinds of graph structures and learning tasks. By drawing a connection between contrastive learning and MP-GNNs, we could leverage these techniques to design better contrastive learning algorithms, as is the goal of this section.

To achieve this, we first establish the basic connections between the basic paradigms in Section 3.1. Built upon these connections, in Section 3.2, we point out the common ideas behind many representative techniques independently proposed by each community in different names. Lastly, we show that we could further borrow some new techniques from the MP-GNN literature for designing new contrastive learning methods in Section 3.3.

## 3.1    CONNECTIONS BETWEEN THE TWO PARADIGMS

Regarding the alignment update and the uniformity update in contrastive learning, we show that they do have a close connection to message passing schemes in graph neural networks.

**Graph Convolution and Alignment Update.** A graph data with $N$ nodes typically consist of two inputs, the input feature $X \in \mathbb{R}^{N \times d}$, and the (normalized) adjacency matrix $\bar{A} \in \mathbb{R}^{n \times n}$. For a feature matrix $H^{(t)} \in \mathbb{R}^{n \times m}$ at the $t$-th layer, the canonical graph convolution scheme in GCN (Kipf & Welling, 2017) gives $H^{(t+1)} = \bar{A}H^{(t)}$. Later, DGC (Wang et al., 2021b) generalizes it to $H^{(t+1)} = (1 - \Delta t)I + \Delta t \bar{A}H^{(t)}$ with a flexible step size $\Delta t$ for finite difference, which reduces the canonical form when $\Delta t = 1$. Comparing it to the alignment update rule in Eq. 4, we can easily notice their equivalence when we take the augmentation graph $\bar{A}$ as the input graph for a GCN (or DGC), and adopt a specific step size $\Delta t = 2\alpha$. In other words, the gradient descent of $F$ using the alignment loss (an optimization step) is equivalent to the graph convolution of $F$ based on the augmentation graph $\bar{A}$ (an architectural module). Their differences in names and forms are only superficial, since inherently, they share the same message passing mechanism.

**Oversmoothing and Feature Collapse.** The equivalence above is the key for us to unlock many intriguing connections between the two seemingly different areas. A particular example is the oversmoothing problem, one of the centric topics in MP-GNNs with tons of discussions, to name a few, Li et al. (2018), Oono & Suzuki (2020), Shi et al. (2022). Oversmoothing refers to the phenomenon that when we apply the graph convolution above for many times, the node features will become indistinguishable from each other and lose discriminative ability. Meanwhile, we have also observed the same phenomenon for the alignment update in Figure 1. Indeed, it is also well known in contrastive learning that with the alignment loss alone, the features will collapse to a single point, known as *complete feature collapse*. Hua et al. (2021) show that when equipped with BN at the output, the features will not fully collapse but still collapse to a low dimensional subspace, named *dimensional collapse*. In fact, both two feature collapse scenarios with the augmentation graph can be characterized using existing oversmoothing analysis of a general graph (Oono & Suzuki, 2020). Therefore, this connection also opens new paths for the theoretical analysis of contrastive learning.

**Self-Attention and Uniformity Update.** Besides the connection in the alignment update, we also notice an interesting connection between the uniformity update rule and the self-attention mechanism.

Table 1: Comparison of linear probing accuracy (%) of contrastive learning methods and their variants inspired by MP-GNNs, evaluated on three datasets, CIFAR-10 (C10), CIFAR-100 (C100), and ImageNet-100 (IN100), and two backbone networks: ResNet-18 (RN18) and ResNet-50 (RN50).

(a) SimSiam and its multi-stage variant (Eq. 13).

| Model | Pretraining | C10 | C100 | IN100 |
|---|---|---|---|---|
| RN18 | SimSiam | 83.8 | 56.3 | 68.8 |
| | + Multi-stage | **84.8** | **58.9** | **70.5** |
| RN50 | SimSiam | 85.9 | 58.4 | 70.9 |
| | + Multi-stage | **87.0** | **59.8** | **72.3** |

(b) SimCLR and its attention variant (Eq. 15).

| Model | Pretraining | C10 | C100 | IN100 |
|---|---|---|---|---|
| RN18 | SimCLR | 84.5 | 56.1 | 62.3 |
| | + Attention | **85.4** | **56.9** | **63.1** |
| RN50 | SimCLR | 88.2 | 59.8 | 66.0 |
| | + Attention | **89.4** | **60.7** | **66.7** |

Self-attention is the key component of Transformer (Vaswani et al., 2017), which recently obtains promising progresses in various scenarios like images, video, text, graph, molecules (Lin et al., 2021). For an input feature $H^{(t)} \in \mathbb{R}^{N \times m}$ at the $t$-th step, the self-attention module gives update:

$$H^{(t+1)} = \bar{A}' H^{(t)}, \text{ where } \bar{A}' = D_{\exp}'^{-1} A_{\exp}', A_{\exp}' = \exp(H^{(t)} H^{(t)\top}). \tag{12}$$

When comparing it to the uniformity update in Eq. 8, we can see that the two rules are equivalent when $\alpha = -1$. Because the step size is negative, self-attention is actually a *gradient ascent* step that will maximize the feature uniformity loss (Eq. 2b). Consequently, stacking self-attention will reduce the uniformity between features and lead to more collapsed representations, which explains the feature collapse behavior observed in Transformers (Shi et al., 2022). Different from previous work (Dong et al., 2021), our analysis above stems from the connection between contrastive learning and MP-GNNs, and it provides a simple explanation of this phenomenon from an optimization perspective.

## 3.2 ANALOGIES IN EXISTING TECHNIQUES

In the discussion above, we establish connections between the two paradigms. Besides, we know that many important techniques and variants have been built upon the basic paradigms in each community. Here, through a unified perspective of the two domains, we spot some interesting connections between techniques developed independently in each domain, while the inherent connections are never revealed. Below, we give two examples to illustrate benefits from these analogies.

**NodeNorm / LayerNorm and $\ell_2$ Normalization.** In contrastive learning, since SimCLR (Chen et al., 2020b), a common practice is to apply $\ell_2$ normalization before calculating the InfoNCE loss, *i.e.*, $f(x)/\|f(x)\|$. As a result, the alignment loss becomes equivalent to the *cosine similarity* between the features of positive pairs. Yet, little was known about the real benefits of $\ell_2$ normalization. As another side of the coin, this node-wise feature normalization technique, in the name of NodeNorm (Zhou et al., 2021), has already been shown as an effective approach to mitigate the performance degradation of deep GCNs by controlling the feature variance. Zhou et al. (2021) also show that LayerNorm (Ba et al., 2016) with both centering and normalization can also obtain similar performance. These discussions in MP-GNNs also help us understand the effectiveness of $\ell_2$ normalization.

**PairNorm / BatchNorm and Feature Centering.** Another common technique in contrastive learning is feature centering, *i.e.*, $f(x) - \mu$, where $\mu$ is the mean of all sample (node) features. In DINO (Caron et al., 2021), it is adopted to prevent feature collapse. Similarly, Hua et al. (2021) also show that BatchNorm (Ioffe & Szegedy, 2015) (with feature centering as a part) can alleviate feature collapse. On the other side, PairNorm (Zhao & Akoglu, 2020) combines feature centering and feature normalization (mentioned above) to alleviate oversmoothing. In particular, they show that PairNorm can preserve the total feature distance of the input (initial features). This, in turn, could be adopted as an explanation for how feature centering works in contrastive learning.

## 3.3 INSPIRED NEW DESIGNS

Besides the analogies drawn above, we find that there are also many other valuable techniques from the MP-GNN literature that have not found their (exact) counterparts in contrastive learning. In this section, we explore two new connections from different aspects: multi-stage aggregation for avoiding feature collapse, and graph attention for better feature alignment.

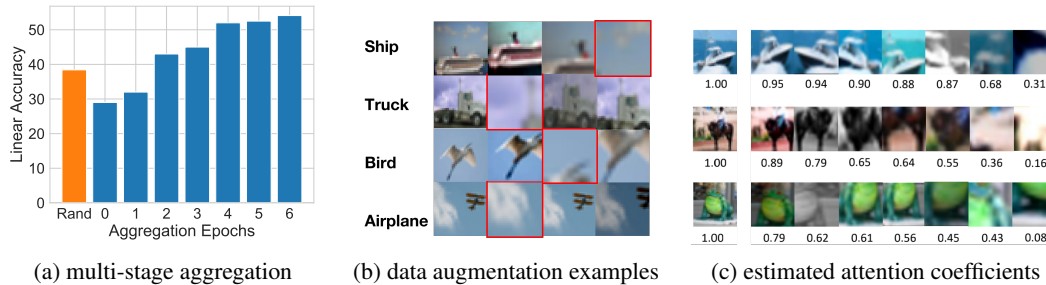

(a) multi-stage aggregation  (b) data augmentation examples  (c) estimated attention coefficients

Figure 2: Experiments on CIFAR-10: (a) Performance of multi-stage aggregation (linear accuracy (%)) and aggregation epochs $s$. (b) natural (1st column) samples and their augmented views (2-4th columns) using SimCLR augmentations. (c) Estimated scores between the anchors (1st column) and the augmented views (2-8th columns) using a pretrained SimCLR model.

### 3.3.1 NEW CONNECTION I: MULTI-STAGE GRAPH AGGREGATION

As discussed above, the alignment loss alone will lead to feature collapse in contrastive learning. Although existing works like BYOL (Grill et al., 2020) and SimSiam (Chen & He, 2021) can alleviate feature collapse via asymmetric architectural designs, we are still not fully clear how they work. Instead, the connection between feature collapse and oversmoothing (Section 3.1) inspires us to borrow techniques from the MP-GNNs to alleviate the feature collapse issue of contrastive learning.

In the MP-GNN literature, a principled solution to oversmoothing is the *jump knowledge* technique proposed in JK-Net (Xu et al., 2018). Instead of adopting features only from the last step $H^{(T)}$ (which might already oversmooth), JK-Net aggregates features from multiple stages, *e.g.,* $H^{(1)}, \ldots, H^{(T)}$, as the final feature for classification. In this way, the knowledge from previous non-over-smoothing steps could directly jump into the final output, which helps improve its discriminative ability. This idea of multi-stage aggregation is also widely adopted in many other MP-GNNs, such as APPNP (Klicpera et al., 2019), GCNII (Chen et al., 2020a), SIGN (Rossi et al., 2020), *etc.*

In this work, we transfer this idea to the contrastive learning scenario, that is, to alleviate feature collapse by combining features from multiple previous epochs. Specifically, we create a memory bank $\mathcal{M}$ to store the latest $s$-epoch features from the same *natural* sample $\bar{x}$. At the $t$-th epoch, for $x$ generated from $\bar{x}$, we have stored old features of it as $z_{\bar{x}}^{(t-1)}, z_{\bar{x}}^{(t-2)}, \ldots, z_{\bar{x}}^{(t-r)}$ where $r = \min(s, t)$. Then we replace the original positive feature $f_\theta(x^+)$ with the aggregation of the old features in the memory bank, where we simply adopt the sum aggregation for simplicity, *i.e.,* $z_{\bar{x}} = \frac{1}{r} \sum_{i=1}^{r} z_{\bar{x}}^{(t-i)}$. Next, we optimize the encoder network $f_\theta$ with the following *multi-stage alignment* loss *alone*,

$$\mathcal{L}_{\text{multi-align}}(\theta) = -\mathbb{E}_{\bar{x}} \mathbb{E}_{x|\bar{x}} (f_\theta(x)^\top z_{\bar{x}}). \tag{13}$$

Afterwards, we will push the newest feature $z = f_\theta(x)$ to the memory bank, and drop the oldest one if exceeded. In this way, we could align the new feature with multiple old features that are less collapsed, which, in turn, also prevents the collapse of the updated features.

**Result Analysis.** From Figure 2a, we can see that when directly applying the multi-stage alignment objective, a larger number of aggregation epochs could indeed improve linear accuracy and avoid full feature collapse. Although the overall accuracy is still relatively low, we can observe a clear benefit of multi-stage aggregation (better than random guess when $s \geq 2$), which was only possible using asymmetric modules, *e.g.,* SimSiam's predictor (Chen & He, 2021). Inspired by this observation, we further combine multi-stage aggregation with SimSiam to further alleviate feature collapse. As shown in Table 1a, the multi-stage mechanism indeed brings cosistent and significant improvements over SimSiam on all three datasets, which clearly demonstrates the benefits of multi-stage aggregation on improving existing non-contrastive methods by further alleviating their feature collapse issues. Experiment details can be found in Appendix C.1.

### 3.3.2 NEW CONNECTION II: GRAPH ATTENTION

A noticeable defect of existing contrastive loss is that they assign an equal weight for different positive pairs. As these positive samples are generated by input-agnostic random data augmentations, there

could be a large difference in semantics between different positive samples. As marked in red (Figure 2b), some augmented views may even lose the center objects, and aligning with these bad cases will bridge samples from different classes together and distort the decision boundary. Therefore, in the alignment update, it is not proper to assign equal weights as in vanilla alignment loss (Eq. 5). Instead, we should adaptively assign different weights to different pairs based on *semantic similarities*.

Revisiting MP-GNNs, we find that there are well-established techniques to address this issue. A well-known extension of graph convolution is *graph attention* proposed in GAT (Veličković et al., 2018), and Transformers can be seen as a special case of GAT applied to a fully connected graph. GAT extends GCN by taking the semantic similarities between neighborhood features (corresponding to positive pairs in the augmentation graph) into consideration. Specifically, for a node $x$ with neighbors $\mathcal{N}_x$, GAT aggregates neighbor features with the attention scores as

$$H_x^{(t+1)} = \sum_{x^+ \in \mathcal{N}_x} \alpha(x, x^+) H_{x^+}^{(t)}, \text{ where } \alpha(x, x^+) = \frac{\exp(e(x, x^+))}{\sum_{x' \in \mathcal{N}_i} \exp(e(x, x'))}. \tag{14}$$

In GAT, they adopt additive attention to compute the so-called attention coefficients $e(x, x')$. Alternatively, we can adopt the dot-product attention as in Transformers, *e.g.*, $e(x, x') = H_x^{(t)\top} H_{x'}^{(t)}$.

Motivated by the design of graph attention, we revise the vanilla alignment loss by reweighting positive pairs with their estimated semantic similarities. Following Transformer, we adopt the dot-product form to estimate the attention coefficient as $e(x, x^+) = f_\theta(x)^\top f_\theta(x^+)$. Indeed, Figure 2c shows that the contrastive encoder can provide decent estimation of semantic similarities between positive samples. Therefore, we adopt this attention coefficient to adaptively reweight the positive pairs, and obtain the following *attention alignment* loss (with the uniformity loss unchanged),

$$\mathcal{L}_{\text{attn}-\text{align}}(\theta) = \frac{1}{2}\mathbb{E}_{x,x^+} \alpha(x, x^+) \|f_\theta(x) - f_\theta(x^+)\|^2. \tag{15}$$

Specifically, we assign the attention score $\alpha(x, x^+)$ as $\alpha(x, x^+) = \frac{\exp(\beta \cdot f_\theta(x)^\top f_\theta(x^+))}{\sum_{x'} \exp(\beta \cdot f_\theta(x)^\top f_\theta(x'))}$, where we simply adopt $M$ negative samples in the mini-batch for computing the normalization, and $\beta$ is a hyperparameter to control the degree of reweighting. We also detach the gradients from the weights as they mainly serve as loss coefficients. The corresponding *attention alignment update* is

$$\text{(local update with attention)} \quad F_x \leftarrow (1 - 2\alpha)F_x + 2\alpha \sum_{x^+}^N \bar{A}_{xx^+} \alpha(x, x^+) F_{x^+}. \tag{16}$$

Compared to the vanilla alignment update (Eq. 5), the attention alignment update adaptively aligns different positive samples according to the *feature-level semantic similarity* $\alpha(x, x^+)$. Thus, our proposed attention alignment can help correct the potential sampling bias of existing data augmentation strategies, and learn more semantic-consistent features.

**Result Analysis.** We follow the basic setup of SimCLR (Chen et al., 2020b) and consider two popular backbones: ResNet-18 and ResNet-50. We consider image classification tasks on CIFAR-10, CIFAR-100 and ImageNet-100. We train 100 epochs on ImageNet-100 and 200 epochs on CIFAR-10 and CIFAR-100. More details can be found in Appendix C.2. As shown in Table 1b, the proposed attention alignment leads to a consistent 1% increase in linear probing accuracy among all settings, which helps verify that our attentional reweighting of the positive samples indeed lead to consistently better performance compared to the vanilla equally averaged alignment loss.

## 4 CONCLUSION

In this paper, we proposed a new message passing perspective for understanding the learning dynamics of contrastive learning. We showed that the alignment update corresponds to a message passing scheme on the augmentation graph, while the uniformity update is a reversed message passing on the affinity graph, and the two message passing rules strike a balance at the equilibrium where the two graphs agree. Based on this perspective as a bridge, we further developed a deep connection between contrastive learning and MP-GNNs. On the one hand, we established formal equivalences between their message passing rules, and pointed out the inherent connections between some existing techniques developed in each domain. On the other hand, borrowing techniques from the MP-GNN literature, we developed two new methods for contrastive learning, and verified their effectiveness on real-world data. Overall, we believe that the discovered connection between contrastive learning and message passing could inspire more studies from both sides to benefit from each other, both theoretically and empirically.

ACKNOWLEDGEMENT

Yisen Wang and Zhouchen Lin were supported by the National Key R&D Program of China (2022ZD0160304), the major key project of PCL, China (No. PCL2021A12), the NSF China (No. 62006153, 62276004), Open Research Projects of Zhejiang Lab (No. 2022RC0AB05), Huawei Technologies Inc., Qualcomm, and Project 2020BD006 supported by PKU-Baidu Fund. Yifei Wang thanks Xiaojun Guo from PKU for valuable feedbacks on an early draft of this work. We thank anonymous reviewers for constructive comments.

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

# A    RELATED WORK

Because our work analyzes contrastive learning in terms of the update rules of the alignment and uniformity losses, it is closely related to Wang & Isola (2020) that firstly propose the alignment-and-uniformity (A+U) perspective of the contrastive learning objective. However, there is a key difference between the two works. In particular, Wang & Isola (2020) only show that if an encoder with perfect A+U *exists*, it would be the minimizer of the InfoNCE loss (see Theorem 1 (Wang & Isola, 2020)). But what we are actually interested is the **opposite**: will the minimization of the InfoNCE loss guide us to the desired A+U properties? This is exactly the focus of our work by studying the learning dynamics of contrastive learning. Besides, we also extend the analysis A+U on the pretraining task (Wang & Isola, 2020) to its theoretical effect on downstream generalization as discussed in Propositions 2. Comparing to Wang et al. (2022), they only *assume* the existence of perfect alignment, while our analysis shows that the contrastive learning process helps improve feature clustering.

Another closely related work is HaoChen et al. (2021) that develops the augmentation graph framework that we build upon. Different from ours, they only characterize the optimal solution of the spectral loss $\mathcal{L}_{\mathrm{sp}}$ directly through eigen-decomposition (ED), which leads to following drawbacks. First, because $N$ can be exponentially large, ED ($\mathcal{O}(N^3)$ complexity) is not directly applicable in practice. Instead, our analysis focuses on the scalable first-order optimization methods that are adopted in practice with good performance in limited training steps. Second, their theory only applies to the spectral loss and fails to characterize the widely adopted InfoNCE loss, which does not admit a closed-form solution. In comparison, our message passing analysis naturally incorporates the InfoNCE loss, as shown in Section 2.2.2.

Besides, Wen & Li (2021) study the contrastive learning process, but only with single-layer networks under toy problems, which is hardly practical. Tian et al. (2021) study the dynamics of non-contrastive learning, but only focus on the predictor parameters. Different from theirs, we adopt a feature space view and analyze the overall evolution of sample features for contrastive learning in practice. Many works before like Wang & Liu (2021) also intuitively discuss the behaviors of contrastive learning gradients. Instead, we provide a formal and systematic study of the learning dynamics of the samples features by characterizing its effect on the downstream performance. In particular, we establish an intriguing connection between contrastive learning and message passing, which is the first time that the two distinctive techniques from different areas are formally bridged together.

# B    PROOF OF MAIN THEORETICAL RESULTS

**Proposition 1.** *When the $x$-th row of $F$ is defined as $F_x = \sqrt{w_x} f_\theta(x)$, we have:*

$$\mathcal{L}_{\mathrm{nce}}(\theta) = \mathcal{L}_{\mathrm{align}}(F_\theta) + \mathcal{L}_{\mathrm{unif}}(F_\theta). \tag{17}$$

*Proof.* We first prove the two equations Eq.(2a) and Eq.(2b), which states the relations between $F$ and $f(x)$.

$$\mathcal{L}_{\mathrm{align}}(F) = \mathrm{Tr}(F^\top L F) = \frac{1}{2}\mathbb{E}_{x,x^+}\|f(x) - f(x^+)\|^2, \tag{2a}$$

$$\mathcal{L}_{\mathrm{unif}}(F) = \mathrm{LSE}(FF^\top) - \|F\|^2 = \mathbb{E}_x \log \mathbb{E}_{x'} \exp(f(x)^\top f(x')) - \mathbb{E}_x\|f(x)\|^2, \tag{2b}$$

For the alignment loss, we have

$$
\begin{aligned}
\mathrm{Tr}(F^\top L F) &= \mathrm{Tr}(F^\top (I - \bar{A}) F) \\
&= -\sum_{x,x'} \sqrt{w_x}\sqrt{w_{x'}} \bar{A}_{xx'} f(x)^\top f(x') + \sum_x w_x f(x)^\top f(x) \\
&= -\sum_x \sum_{x'} A_{xx'} f(x)^\top f(x') + \sum_x \sum_{x'} A_{xx'} f(x)^\top f(x) \\
&= -\sum_x \sum_{x'} A_{xx'} f(x)^\top f(x') + \frac{1}{2}\sum_x \sum_{x'} (A_{xx'} f(x)^\top f(x) + A_{x'x} f(x')^\top f(x')) \\
&= -\sum_x \sum_{x'} A_{xx'} f(x)^\top f(x') + \frac{1}{2}\sum_x \sum_{x'} (A_{xx'} f(x)^\top f(x) + A_{xx'} f(x')^\top f(x')) \\
&= \frac{1}{2}\sum_x \sum_{x'} A_{xx'} \|f(x) - f(x')\|^2 \\
&= \frac{1}{2}\mathbb{E}_{x,x^+} \|f(x) - f(x^+)\|^2
\end{aligned}
\tag{19}
$$

For the uniformity loss, we have

$$
\begin{aligned}
\mathrm{LSE}(FF^\top) - \|F\|^2 &= \mathrm{Tr}(D \log(D \deg(\exp(D^{-1/2} FF^\top D^{-1/2})))) - \|F\|^2 \\
&= \sum_x w_x \log \sum_{x'} w_{x'} \exp(f(x)^\top f(x')) - \sum_x w_x \|f(x)\|^2 \\
&= \mathbb{E}_x \log \mathbb{E}_{x'} \exp(f(x)^\top f(x')) - \mathbb{E}_x \|f(x)\|^2.
\end{aligned}
\tag{20}
$$

Therefore, we have

$$
\begin{aligned}
\mathcal{L}_{\mathrm{nce}}(\theta) &= -\mathbb{E}_{x,x^+}[f(x)^\top f(x^+)] + \mathbb{E}_x \log \mathbb{E}_{x'}[\exp(f(x)^\top f(x'))] \\
&= -\mathbb{E}_{x,x^+}[f(x)^\top f(x^+)] + \mathbb{E}_x(f(x)^\top f(x)) + \mathbb{E}_x \log \mathbb{E}_{x'}[\exp(f(x)^\top f(x'))] - \mathbb{E}_x \|f(x)\|^2 \\
&= \frac{1}{2}\mathbb{E}_{x,x^+} \|f(x) - f(x^+)\|^2 + \mathbb{E}_x \log \mathbb{E}_{x'}[\exp(f(x)^\top f(x'))] - \mathbb{E}_x \|f(x)\|^2 \\
&= \mathcal{L}_{\mathrm{align}}(F_\theta) + \mathcal{L}_{\mathrm{unif}}(F_\theta).
\end{aligned}
\tag{21}
$$
$\square$

**Proposition 2.** *Denote the distance between a feature matrix $X$ and a subspace $\Omega$ by $d_\Omega(X) := \inf\{\|X - Y\|_F \mid Y \in \Omega\}$. Assume that data augmentations are label-preserving. Then, for samples in each class $k$, denoting their adjacency matrix as $\bar{A}_k$ and their features at $t$-th step as $F_k^{(t)}$, the message passing in Eq. 4 leads to a smaller or equal distance to an one-dimensional subspace $\mathcal{M}$*

$$
d_{\mathcal{M}}(F_k^{(t+1)}) \leq |1 - 2\alpha\lambda_{\mathcal{G}_k}| \cdot d_{\mathcal{M}}(F_k^{(t)}),
\tag{22}
$$

*where $\lambda_{\mathcal{G}_k} \in [0, 2]$ denotes the intra-class algebraic connectivity of $\mathcal{G}$ (Chung, 1997), and $\mathcal{M} = \{e_1 \otimes y | y \in \mathbb{R}^c\}$, and $e_1$ is the eigenvector corresponding to the largest eigenvalue of $\bar{A}_k$.*

*Proof.* Under the condition that the data augmentations are label-preserving, there are no edges between inter-class nodes. Thus, the propagation of the alignment update only happens within each class. We denote the propagation matrix as $Q_k = (1-2\alpha)I + 2\alpha\bar{A}_k$. Then we have $F_k^{(t+1)} = Q_k F_k^{(t)}$. Since $\bar{A}_k$ is a symmetrically normalized adjacent matrix, we can denote the spectrum of $\bar{A}_k$ as $1 = \lambda_1 \geq \cdots \geq \lambda_n \geq -1$. Then the spectrum of $Q_k$ is $q_i = 1 - 2\alpha + 2\alpha\lambda_i, i = 1, 2, \ldots, N$, and specifically, $q_1 = 1$. We denote the corresponding orthonormal basis of $Q_k$ as $[e_1, \ldots, e_n]$ since $Q_k$ is a symmetric matrix.

Accordingly, we could decompose $F_k^{(t)}$ according to the orthonormal basis as

$$
F_k^{(t)} = \sum_{i=1}^n e_i \otimes h_i, h_i \in \mathbb{R}^m.
\tag{23}
$$

Therefore, the projection of $F_k^{(t)}$ to the space $\mathcal{M} = \{e_1 \otimes y | y \in \mathbb{R}^c\}$ gives the first component of the decomposition (Eq.23), *i.e.*, $e_1 \otimes h_1$, and the residual is the remaining components orthogonal to $e_1$, *i.e.*, $\sum_{i=2}^n e_i \otimes h_i$, with the following distance,

$$d_{\mathcal{M}}^2(F_k^{(t)}) = \left\| \sum_{i=2}^n e_i \otimes h_i, h_i \in \mathbb{R}^m \right\|_F^2 = \sum_{i=2}^n \|h_i\|_2^2. \tag{24}$$

Similarly, we can also deduce the distance of the updated features. Since $Qe_i = q_i e_i$, we have

$$QF_k^{(t)} = e_1 \otimes h_1 + \sum_{i=2}^n e_i \otimes (q_i h_i). \tag{25}$$

Thus, its projection to $\mathcal{M}$ is also $e_1 \otimes h_1$, and we have

$$d_{\mathcal{M}}^2(QF_k^{(t)}) = \left\| \sum_{i=2}^n e_i \otimes (q_i h_i) \right\|_F^2 = \sum_{i=2}^n \|q_i h_i\|_2^2. \tag{26}$$

Combining Eqs.24,26, we arrive at

$$d_{\mathcal{M}}^2(F_k^{(t+1)}) = d_{\mathcal{M}}^2(QF_k^{(t)}) = \sum_{i=2}^n \|q_i h_i\|_2^2 \le \sum_{i=2}^n q_2^2 \|h_i\|_2^2 = q_2^2 d_{\mathcal{M}}^2(F_k^{(t)}). \tag{27}$$

Note that $1 - \lambda_2$ is just the algebraic connectivity of $\mathcal{G}$. Therefore, with $q_2 = 1 - 2\alpha + 2\lambda_2$, we have

$$d_{\mathcal{M}}^2(F_k^{(t+1)}) \le (1 - 2\alpha \lambda_{\mathcal{G}}^{in})^2 d_{\mathcal{M}}^2(F_k^{(t)}) \le d_{\mathcal{M}}^2(F_k^{(t)}), \tag{28}$$

By taking square root on this inequality, we arrive at the final result

$$d_{\mathcal{M}}(F_k^{(t+1)}) \le |(1 - 2\alpha \lambda_{\mathcal{G}}^{in})| \cdot d_{\mathcal{M}}(F_k^{(t)}) \le d_{\mathcal{M}}(F_k^{(t)}), \tag{29}$$

which completes the proof. $\qquad \square$

**Proposition 3.** *The contrastive learning dynamics with the InfoNCE loss saturates when the real and the fake adjacency matrices agree, i.e., $\bar{A} = \bar{A}'_{\text{nce}}$, which means that element-wisely, we have:*

$$\forall\, x, x^+ \in \mathcal{X}, \ \ \mathcal{P}_d(x^+|x) = \mathcal{P}_\theta(x^+|x) \triangleq \frac{\exp(f_\theta(x)^\top f_\theta(x^+))}{\sum_{x'} \exp(f_\theta(x)^\top f_\theta(x'))}. \tag{30}$$

*That is, the ground-truth data conditional distribution $\mathcal{P}_d(x^+|x)$ equals to the Boltzmann distribution $\mathcal{P}_\theta(x^+|x)$ estimated from the encoder features.*

*Proof.* Notice that the contrastive update takes the form of

$$F^{(t+1)} = F^{(t)} - \alpha \nabla_{F^{(t)}} \mathcal{L}_{\text{nce}}(F^{(t)}) = F^{(t)} + 2\alpha(\bar{A} - \bar{A}'_{\text{nce}})F^{(t)}. \tag{31}$$

Therefore, when $\bar{A} = \bar{A}'_{\text{nce}}$, we have $F^{(t+1)} = F^{(t)}$, which means that the contrastive learning dynamics with the InfoNCE loss saturates. At this time, we have

$$\bar{A}(i,j) = \bar{A}_{\text{nce}}(i,j) = \frac{\exp(f(x)^\top f(x'))}{\sum_k \exp(f(x)^\top f(x_k))}, \tag{32}$$

which implies the result in the proposition. $\qquad \square$

## C  Experimental Details

### C.1  Details of Multi-stage Graph Aggregation

For the experiments in Figure 2a, we follow the default designs and settings of SimSiam (Chen & He, 2021) on CIFAR-10, except that we remove its predictor to focus on the effect of multi-stage aggregation. We adopt ResNet-50 as the backbone and pretrain for 100 epochs. During the training process, we use a memory bank to store the features of all samples from the last $k$ epochs. To obtain the final aggregated feature, we average the features from the last $k$ epochs. Afterwards, we update

the memory bank with the feature of the positive sample in the current epoch, and discard old features if they are not from the last $k$ epochs.

For the experiments in Table 1a, we also follow SimSiam's setting and consider image classification tasks on three benchmark datasets: CIFAR-10, CIFAR-100 and ImageNet-100. We train ResNet-18 and ResNet-50 backbones for 200 epochs with the default hyperparameters of SimSiam (Chen & He, 2021). Specifically, we use the SGD optimizer with 256 batch size and 5e-5 weight decay. To implement the multi-stage variant of SimSiam, after 50 warmup epochs, we replace the target feature in SimSiam (which is orginally the representation of a positive sample) with the multi-stage aggeragated features over the last 2 epochs (current epoch and the last epoch).

## C.2 DETAILS OF GRAPH ATTENTION EXPERIMENTS

We use SimCLR (Chen et al., 2020b) as our baseline and ResNet-18,ResNet-50 as the backbones. We follow the backbone with a projection MLP to map the features to the 128-dimensional projection space. We consider image classification tasks on CIFAR-10, CIFAR-100 and ImageNet-100. There are two stages in contrastive learning, *i.e.,* self-supervised pretraining and linear evaluation. We pretrain the encoder for 100 epochs on ImageNet-100 and for 200 epochs on CIFAR-10 and CIFAR-100. We use the LARS optimizer with cosine annealed learning rate schedule and 512 batch size. After the pretraining process, we train a linear classifier following the frozen backbones using the SGD optimizer. To obtain a more accurate estimation of semantic similarity, we use the features before the projection layer and warmup the model for 30 epochs.

