# OpenReview forum: "A Message Passing Perspective on Learning Dynamics of Contrastive Learning"
_ICLR.cc/2023/Conference — ICLR 2023 poster_

### Official Review · Reviewer_7ZHM · 2022-10-22

**Confidence:** 3
**Correctness:** 4
**Technical Novelty And Significance:** 3
**Empirical Novelty And Significance:** 4
**Recommendation:** 8

**Clarity, Quality, Novelty And Reproducibility:**

This paper is well-written and technically sound. Detailed proofs are attached in the appendix. Basically, this paper makes a novel interpretation of contrastive learning which is very interesting. The experimental details are also provided which enhances its reproducibility.

**Strength And Weaknesses:**

Strengths:
1. This paper studies a significant problem about how data augmentation affects the learning dynamics of contrastive learning, although previous works have made some trials, this paper has a totally new perspective that is more interpretable.
2. The analogy from MP-GNN is very interesting, such as the oversmoothing problem and feature collapse, this is inspiring for the future study of contrastive learning.
3. It proposes some new designs which are also justified by the experimental results, the GAT-like attentive score weights is reasonable and promising.

Weaknesses:
1. The analogies and two new designs mainly focus on the alignment updates, while negative sampling for the uniformity updates is a well-known problem. I think the authors should also pay attention to the negative sampling part such as the hard negative sampling [1].


[1] Contrastive Learning with Hard Negative Samples. Robinson et al., ICLR 2021.


**Summary Of The Paper:**

This paper casts a new understanding of contrastive learning, which interprets the alignment update and uniformity update as message-passing steps and bridges contrastive learning and message passing. It interprets some techniques used in contrastive learning from the perspective of message passing neural network and proposes some inspiration for the modification of contrastive learning, some empirical results also justify these changes inspired by MPNN.

**Summary Of The Review:**

This paper is well-written and presents an innovative understanding of contrastive learning. It connects contrastive learning with MP-GNN in both theory and the analogy of problems/techniques. It also provides some ideas for new designs in contrastive learning inspired by the existing tools in MP-GNN. I think this is a solid paper and vote for accept.

---

> ### Author Response · Authors · 2022-11-13
> **Response to Reviewer 7ZHM**
>
>
> We thank Reviewer 7HZM for your encouraging comments appreciating the significance, soundness, and reproducibility of our work.
> Below, we address your main concerns.
>
> ---
> **Q1.** The analogies and two new designs mainly focus on the alignment update. I think the authors should also pay attention to the negative sampling part such as the hard negative sampling [1].
>
> **A1.** The main methodology of our work is to leverage the connections between contrastive learning and existing message-passing GNNs (MP-GNNs) to devise effective CL methods. However, up to our knowledge, we find few counterparts in MP-GNNs to negative sampling techniques in contrastive learning. This is because most existing MP-GNNs still focus on designing message-passing rules between **the neighbors given *a priori* (i.e., the positive samples in contrastive learning)**. A few works [2,3] explore the **graph rewiring** technique that re-defines neighbors according to their feature similarity (similar to the affinity graph $A'$), but no one uses these rewired neighbors to promote feature uniformity as in contrastive learning. Therefore, the lack of discussion of negative samples is mainly due to the lack of proper counterparts in existing MP-GNNs. Nevertheless, we believe this is an important direction for future work.
>
> [1] Contrastive Learning with Hard Negative Samples. Robinson et al., ICLR 2021.
>
> [2] Rewiring with Positional Encodings for Graph Neural Networks. Brüel-Gabrielsson et al. https://arxiv.org/pdf/2201.12674.pdf
>
> [3] Understanding over-squashing and bottlenecks on graphs via curvature. Topping et al. ICLR 2022.  https://arxiv.org/pdf/2111.14522.pdf
>
> ---
> Thanks again for your careful reading and for appreciating our work. Hope the elaboration above could address your concerns. Please let us know if you have further questions.

---

### Official Review · Reviewer_WHor · 2022-10-25

**Confidence:** 4
**Correctness:** 3
**Technical Novelty And Significance:** 3
**Empirical Novelty And Significance:** 2
**Recommendation:** 6

**Clarity, Quality, Novelty And Reproducibility:**

The connection between contrastive learning and message passing GNN methods is novel and potentially useful, to the best of my knowledge. Most of the paper is clearly written and it was easy to follow even in 1 pass. Some typos and bugs in equations/derivations have been pointed out above.

**Strength And Weaknesses:**

**Strengths**

- The paper establishes many connections between contrastive learning updates and message passing methods that could potentially help both fields from advancements in the other.
- It is also reasonably easy to follow and well written (comments towards the end).
- The proposed method that utilizes graph attention mechanism perform slightly better than vanilla CL

**Weaknesses**

- While the paper makes many connections, most are derived from pretty straightforward or known calculations (the connections can still be valuable). If not technical, ideally the main contribution could have been to leverage the connections for new contrastive learning algorithms that do well. The only one that gets good performance (and improvements) is the graph attention idea.

- Using multi-stage aggregation, i.e. using features from multiple epochs rather than just the current one, does avoid complete feature collapse without negatives. However the final performance is still far from CL; I think ~45% on CIFAR-10 is achievable with even a randomly initialized network, so I'm not sure how much to read into those results of ~50%.

- Typos/bugs in derivations and proofs (that may be fixable)
  - I fail to see why $LSE(FF^{\top})$ is the same as $\mathbb{E}\_{x} \log(\mathbb{E}\_{x'} \exp(f(x)^{\top}f(x^{+})))$ in Eq. (2b). It seems like it should be something like $Tr(D \log(deg(D \exp(D^{-1/2}FF^{\top} D^{-1/2}))))$. The proof in Eq (20) seems wrong because the $w$ term vanishes from the $\exp$ in the next step. Similar the expression for $A'\_{\text{exp}}$ below Eq (7) seems wrong and other places that use $FF^{\top}$ kind of term by ignoring the $D$ factors. I'm not entirely sure if this affects other results (seems like it won't for uniform distribution), but it would be nice to get some clarification from the authors about this
  - In Eq (8) should the first term be $(1+2\alpha) I$ instead of $(1+\alpha)I$? An so should Eq (9) not have the $(1-\alpha)$ term? If not, then the Equilibrium distribution calculation will be incorrect, since $\bar{A} = A'$ will give $F^{(t+1)} = (1-\alpha) F^{(t)}$.

- It might be worth mentioning in Proposition 3 that the equilibrium distribution is true only if $f$ is allowed to be arbitrarily expressive and that it ignores sample complexity considerations. Also, such a result of distribution recovery has been shown in prior work using different techniques, Theorem 1 in [1].

- Other comments/questions
  - Missing citation for GAT on page 1
  - "Empirically, we show that both techniques leads to clear benefits on benchmark datasets": I would not say that there are clear benefits for both techniques


[1] Zimmerman et al. Contrastive Learning Inverts the Data Generating Process

**Summary Of The Paper:**

This paper studies contrastive learning (CL) by looking the dynamics of (unconstrained) gradient descent for the contrastive loss and connecting it to message passing schemes on the augmentation graph.

- The "alignment update" for CL is a message passing scheme on the augmentation graph, because each feature is updated as a weighted sum of neighborhood features in the augmentation graph
- The "uniformity update" is equivalent to message passing on a "fake affinity graph" that is itself defined by the similarities induces by the current features.
- Under this view, the contrastive update is a competition between two message passing rules and this helps characterize the "equilibrium solution"

It further elucidates connections/analogies between various techniques in contrastive learning and message passing methods on graph neural networks (MP-GNNs), like layer/batch normalization, feature collapse etc.
Inspired by these connections, the paper designs new contrastive learning variants that borrow techniques from GNNs like graph attention jumpy knowledge etc. that perform reasonably well on standard benchmarks.

- Using graph attention mechanism for adaptive positives with a weighted alignment term $\alpha_{f}(x, x^+) f(x)^{\top} f(x^+)$, to avoid considering far off positives. Leads to ~1% improvement over vanilla evaluation.

**Summary Of The Review:**

I think the connection between CL and GNNs could open up new avenues for ideas in contrastive learning. This paper showed one such connection through graph attentions that lead to a small benefit. Overall it would have nicer if more such connections/insights arose. I believe that the overall contribution of the paper is positive. However due to bugs in the derivations and proofs (listed above), that may or may not affect some results, I am hesitant to accept the paper immediately, and have thus assigned a score of weak reject. Happy to raise the score after further clarification from the authors.

---

> ### Author Response · Authors · 2022-11-13
> **Response to Reviewer WHor (2/2)**
>
> **Q3.** Typos / bugs in derivations and proofs.
>
> **A3.** Thanks for your detailed reading! We have fixed the typos and bugs you pointed out in the revision following your suggestions.
>
> > a) Why $LSE(FF^{\top})$ is the same as $E_x \log(E_{x'} \exp(f(x)^\top f(x^{+})))$ in Eq. (2b). The proof in Eq (20) seems wrong because the $w$ term vanishes from the $w$ in the next step.
>
> Indeed, there is a typo here, as we forgot the $w_x$ term in $F_x$ when reformuating. As you suggested, it should be $LSE(FF^{\top})=Tr(D \log(\operatorname{deg}(D \exp(D^{-1/2}FF^{\top} D^{-1/2}))))$. We have fixed it and revised the proof (Eq. 20) accordingly in revision.
> > b) Similar the expression for $A'_{\text{exp}}$ below Eq (7) seems wrong and other places that use $FF^\top$ kind of term by ignoring the $D$ factors. I'm not entirely sure if this affects other results (seems like it won't for uniform distribution), but it would be nice to get some clarification from the authors about this.
>
> Similarly, we should put $D$ back, and $A'$ should be $D^{-1/2}FF^{\top} D^{-1/2}$. As it is a simply a reformulation of $f(x)^\top f(x')$, as you observed, this typo does not affect other results under the uniform data distribution. We have it fixed in the revision.
>
> > c) In Eq (8) should the first term be $(1+2\alpha) I$ instead of $(1+\alpha) I$?
>
> It is indeed $(1+\alpha) I$. Different from Eq. (7), in Eq. (8), we apply stop gradient to the target branch, which affects both terms in $\mathcal{G}_{\rm unif}(F)=\operatorname{LSE}(FF^\top)-\|F\|^2$. Specifically, with stop gradient, the gradient of $-\|F\|^2=-\operatorname{tr}(F^\top F)$ is $-F$, because one of $F$ is constant. As a result, we have $(1+\alpha)F$ in Eq. (8).
>
> > d) And so should Eq (9) not have the $(1-\alpha)$ term? If not, then the Equilibrium distribution calculation will be incorrect.
>
> Indeed, we also made a typo here. Unifying Eq. (8) and (9), the combined update should be
> $$F^{(t+1)}=F^{(t)}+\alpha(\bar A-\bar A')F^{(t)}.$$
> We have now fixed it in the revision.
>
> Thanks for your careful reading and valuable suggestions! We have fixed all the typos and bugs that you have mentioned above. Hope you find it resolving your concerns.
>
> ---
>
> **Q4.** It might be worth mentioning in Proposition 3 that the equilibrium distribution is true only if $f$ is allowed to be arbitrarily expressive and that it ignores sample complexity considerations. Also, such a result of distribution recovery has been shown in prior work using different techniques, Theorem 1 in [1].
>
> **A4.** Actually, we have mentioned this property of $f$ in the problem setup (Sec 2.1), where we state that as over-parameterized neural networks are **very expressive and adaptive**, we assume the feature matrix $F_\theta$ to be **unconstrained ignoring the dependence on parameter and can be updated freely**. Following your suggestions, we further mention that we also omit sample complexity in the revision. We note that this assumption is also adopted in Haochen et al (2021) and many other DL theories, such as, layer peeled models.
>
> >  Also, such a result of distribution recovery has been shown in prior work using different techniques, Theorem 1 in [1].
>
> Indeed, similar to our Proposition 3, Zimmerman et al. [1] also show that contrastive learning can recover the data distribution from a nonlinear ICA perspective. The main difference is that they derive this result based on the learning objective, while ours focuses on how the learning dynamics attains this equilibrium. Besides, to obtain this distribution recovery property, Zimmerman et al. [1] rely heavily on the data generation process assumption, while ours does not. We have added this discussion to **Related Work (Appendix C)**.
>
> [1] Zimmerman et al. Contrastive Learning Inverts the Data Generating Process
>
> ---
>
> **Q5.** Other comments/questions.
>
> **A5.**
>
> > a) Missing citation for GAT on page 1
>
> Thanks and we have fixed it in the revision.
>
> > b) "Empirically, we show that both techniques leads to clear benefits on benchmark datasets": I would not say that there are clear benefits for both techniques.
>
> In **A1**, we have elaborated the significance of the multi-stage aggregation experiment, and in **A2**, we further add a new experiment in **Appendix D** showing that it can also bring clear benefits on benchmark datasets. Thus, both techniques proposed in our work are shown useful in practice.
>
> ---
>
> Thanks again for your valuable and constructive comments, and hope our revisions and explanations above address your concerns. We are looking forward to your reply and please let us know if you have any further questions.

---

> ### Author Response · Authors · 2022-11-13
> **Response to Reviewer WHor (1/2)**
>
>
> We thank Reviewer WHor for your careful reading and for appreciating the novelty and clarity of this work. We have fixed the minor points you mentioned in the revised paper. Below, we address your main concerns.
>
> ---
> **Q1.** While the paper makes many connections, most are derived from pretty straightforward or known calculations (the connections can still be valuable).
>
> **A1.** We believe that the simplicity of our derivation might be a bit hindsight, given the fact that we have already properly reformulated the contrastive learning objectives. From our view, the main theoretical contribution of this work is **a new theoretical framework to understand and characterize contrastive learning process**, and we achieve this with a new reformulation to ensure that the alignment and uniformity updates could nicely correspond to message passing formulas (**Prop 1**).
>
> Besides, we also made non-trivial technical discussions from our message passing perspective:
> - In **Prop 2**, we firstly pointed out that how alignment update affects downstream performance depends quantitatively on the algebraic connectivity of the augmentation graph;
> - In **Prop 3**, we revealed how contrastive learning recovers the data distribution through analysis of the equilibrium states;
> - In **Sec 3.1 (Self-Attention and Uniformity Update)**, we provide a new theoretical explanation for the oversmoothing phenomenon of self-attention modules from an optimization perspective.
>
> These theoretical discussions we established also bring new insights for understanding contrastive learning and self-attention modules.
>
> ---
> **Q2.** The only one that gets good performance (and improvements) is the graph attention idea. However, the performance of multi-stage aggregation is still far from CL; I think ~45% on CIFAR-10 is achievable with even a randomly initialized network.
>
> **A2.** Following your suggestions, we also evaluate the performance of a randomly initialized encoder, and add it to Figure 2a (results quoted below).
>
> | Method | Acc |
> | -- | --|
> | Random Init | 38.4 |
> | No Aggregation | 29.1 |
> | Aggr (1 epoch) | 32.3 |
> | Aggr (3 epochs) | 42.7 |
> | Aggr (6 epochs) | **54.1** |
>
> We can see that multi-stage aggregation **(54.1%)** indeed shows significant improvement over both the collapsed baseline (29.1%), but **also over the random init (38.4%)**, implying that **multi-stage aggregation alone indeed learns useful features**. This experiment shows that we can non-trivially avoid feature collapse without introducing asymmetric modules, which is a new message worthy of noting to the community.
>
> **Clear benefits of multi-stage aggregation**. Since multi-stage aggregation alone can help alleviate feature collapse, we further show that it can yield clear benefits when combined with existing techniques for avoiding feature collapse, such as SimSiam. We compare SimSiam and its multi-stage variant on CIFAR-10 with 200-epoch training of ResNet-18. The results are shown below.
>
> | Method | Acc (%) |
> | -- | -- |
> |SimSiam | 83.82 |
> | SimSiam + multi-stage (ours) | **84.75 (+0.93)** |
>
> From the table above, we can see that combining SimSiam with multi-stage can indeed bring clear improvements by further alleviating feature collapse. This shows that the multi-stage aggregation derived from our message passing perspective can indeed bring clear benefits to existing self-supervised methods.
>
> We have added these new experiments in **Appendix D**.

---

> ### Comment · Reviewer_WHor · 2022-11-19
> **Response to authors**
>
> Thank you for the detailed responses and the new experiments. The responses, bug fixes, and the new experiment with the random init baseline and multi-stage aggregation + SimSiam seem to sufficiently clarify most of my concerns. I will thus be increasing my score in the accept range.
>
> A quick note on Zimmerman et al.: It is a good point that the current paper shows the distribution recovery using dynamics. Although if I understand the results in Zimmerman et al. correctly, they do not need the data generation process for the distribution recovery, but only for the subsequent arguments.

---

> > ### Author Response · Authors · 2022-11-19
> > **Thanks and More Results on Multi-stage Aggregation**
> >
> > Thanks for your reply and for appreciating our responses! We are delighted to hear that we have addressed your concerns, particularly the SimSiam + multi-stage experiments.
> >
> > **More complete on multi-stage aggregation.** With more available time, we also experiment the multi-stage variant of SimSiam on more datasets (CIFAR-100, ImageNet-100), as shown below (added to **Table 2 in Appendix D**).
> >
> > | Dataset | Method | Top-1 Acc (%) |
> > | --- | --- | --- |
> > | CIFAR-10 | SimSiam | 83.82 |
> > |  | SimSiam-MultiStage (ours) | **84.75 (+0.93)** |
> > | CIFAR-100 | SimSiam | 56.34 |
> > |  | SimSiam-MultiStage (ours) | **58.87 (+2.53)** |
> > | ImageNet-100 | SimSiam | 68.76 |
> > |  | SimSiam-MultiStage (ours) | **70.52 (+1.76)** |
> >
> > From the table above, we can see that the multi-stage aggregation can lead to **consistent and clear improvements over multiple benchmark datasets**. More details can be found in **Appendix D** in the newly revised version.
> >
> > **On Zimmerman et al.** As far as we could see, their distribution recovery result (Prop 1) is established over the latent space, between $p(\tilde{z}|z)$ and $q(\tilde z|z)$. To derive this result, in the proof of both Theorem 1 and Prop 1, they relied on the assumption of the data generation process in Eq. 2. Therefore, it seems that they still require the overall assumptions on the data generation process. Please let us know if there is something we miss here.
> >
> > Thanks again for your reply and hope the results above could further ease your concerns.

---

### Official Review · Reviewer_KQoy · 2022-10-27

**Confidence:** 4
**Correctness:** 4
**Technical Novelty And Significance:** 4
**Empirical Novelty And Significance:** Not applicable
**Recommendation:** 8

**Clarity, Quality, Novelty And Reproducibility:**

This paper is clearly written. I do not spot any obvious errors. I think this paper makes a novel contribution both theoretically and empirically for understanding and improving contrastive learning. The authors do not provide source code so I cannot comment on the reproducibilty.

**Strength And Weaknesses:**

Strengths:
+ This paper presents a very interesting perspective on understanding contrastive learning by formulating alignment and uniformity as message passing on graphs
+ This paper is well-written and the idea is easy to follow
+ The message passing formulation is supported by both theoretical analysis and empirical evidence

Weaknesses:
I only have several minor points:
- The second paragraph in the introduction section is a little bit confusing. I would like to suggest the authors to elaborate on "model distribution" and "data distribution".
- I wonder how we can understand the MoCo approach that maintains a memory bank for negatives by constructing similar message passing graphs?
- In Page 1, GAT is not cited correctly.
- In Page 5, reference (Eq. 30) should be (Eq. 11).


**Summary Of The Paper:**

This paper revisits contrastive learning objectives from the feature propagation perspective. Specifically, by casting alignment and uniformity as two message propagation procedures on two respectively graphs, we can establish equivalence between contrastive learning and message passing graph neural networks. In this way, we can inherit existing techniques in graph neural networks to improve contrastive learning performance.

**Summary Of The Review:**

I like the paper very much. I think the authors make a nontrivial contribution in understanding and improving contrastive learning.

---

> ### Author Response · Authors · 2022-11-13
> **Response to Reviewer KQoy**
>
>
> We thank Reviewer KQoy for appreciating the novelty and solidness of our work. Below, we address your main concerns.
>
> ---
>
> **Q1.** The second paragraph in the introduction section is a little bit confusing. I would like to suggest the authors to elaborate on "model distribution" and "data distribution".
>
> **A1.** Thanks for your suggestions. Here, we are referring to the balance is striked when the learned distribution $\mathcal{P}_\theta$ (model distribution) matches the ground-truth data distribution $\mathcal{P}_d$ (Proposition 3). We have clarified these concepts in the revision.
>
> ---
>
> **Q2.** I wonder how we can understand the MoCo approach that maintains a memory bank for negatives by constructing similar message passing graphs?
>
> **A2.** In this work, we mainly study the population version of contrastive loss w.r.t. all samples in the input space $\mathcal{X}$. Therefore, **the message passing graphs model the relationship between all samples in $\mathcal{X}$**, and the update rule also includes all samples, and specifically, all the positive and negative samples for each sample.
>
> In practice, we cannot compute all negative samples due to efficiency issues, and SimCLR and MoCo offer **two different strategies to subsample the negative samples for approximating the uniformity update**: SimCLR based on mini-batch samples, and MoCo based on large memory bank and exponential moving average. Comparing the two, SimCLR' sampling is less unbiased but requires a large batch size for accurate estimation, while **MoCo's memory bank estimate has smaller variance and allow small batch sizes, but may be potentially biased**.
>
> Therefore, our message passing perspective also provides a unified perspective for analysing different contrastive learning methods.
>
> ---
> **Q3.** In Page 1, GAT is not cited correctly. In Page 5, reference (Eq. 30) should be (Eq. 11).
>
> **A3.** Thanks for pointing them out, and we have them fixed in the revision.
>
> ---
> Thanks again for appreciating our work and for your constructive comments. Please let us know if you have further questions. We would be happy to take them during the rebuttal stage.

---

### Decision · Program_Chairs · 2023-01-20

**Decision:**

Accept: poster

**Justification For Why Not Higher Score:**

Overall, the paper makes a positive and significant contribution to the field of contrastive learning and graph neural networks, and deserves to be accepted as a poster. The paper is not strong enough for an spotlight presentation, as it does not provide a major breakthrough or a comprehensive evaluation of the proposed methods. It also does not fully explore the implications of the message passing perspective for the negative sampling and uniformity update, which are important aspects of contrastive learning. The paper could be further improved by clarifying some of the claims and assumptions, fixing the errors in the derivations and proofs, and discussing more related work and limitations.

**Justification For Why Not Lower Score:**

The paper should not be rejected because it presents a novel and insightful view of contrastive learning, which could inspire future research and applications. It also provides some evidence of the benefits of the proposed methods, which are based on solid theoretical foundations. The paper is well-organized and easy to follow, and the authors provide sufficient details and proofs in the appendix. The paper does not have any major flaws or ethical issues that would warrant a rejection.

**Metareview: Summary, Strengths And Weaknesses:**

This paper proposes a novel perspective on contrastive learning by relating it to message passing on graphs. It shows that the alignment and uniformity updates in contrastive learning can be seen as message passing schemes on two different graphs, one induced by data augmentation and the other by feature similarity. It also draws connections and analogies between various techniques and challenges in contrastive learning and message passing graph neural networks, such as layer normalization, feature collapse, and oversmoothing. Based on these insights, the paper introduces two new variants of contrastive learning that leverage graph attention and multi-stage aggregation, and demonstrates their effectiveness on standard benchmarks.

The reviewers generally appreciate the paper's contributions and find it well-written and technically sound. They agree that the paper establishes interesting and potentially useful connections between contrastive learning and message passing, and that the proposed methods are novel and empirically validated. They also point out some minor issues, such as typos, missing citations, and unclear statements. Reviewer 2 also raises some concerns about the correctness of some derivations and proofs, and the significance of some results. The authors are expected to address these issues in the final version of the paper.

**Note From Pc:**

if the above contains the word "oral" or "spotlight" please see: "oral" presentation means -> notable-top-5% and "spotlight" means -> notable-top-25%. As stated in our emails, we are disassociating presentation type from AC recommendations